# Electronic Health Records, Interoperability and Patient Safety in Health Systems of High-income Countries: A Systematic Review Protocol

Edmond Li [ORCID] ,[1] Jonathan Clarke [ORCID] ,[1,2] Ana Luisa Neves [ORCID] ,[1,3] Hutan Ashrafian [ORCID] ,[1] Ara Darzi[1]

For numbered affiliations see end of article.

**Correspondence to**
Dr Edmond Li;
edmond.li19@imperial.ac.uk

## ABSTRACT

**Introduction** The availability and routine use of electronic health records (EHRs) have become commonplace in healthcare systems of many high-income countries. While there is an ever-growing body of literature pertaining to their use, evidence surrounding the importance of EHR interoperability and its impact on patient safety remains less clear. There is, therefore, a need and opportunity to evaluate the evidence available regarding this relationship so as to better inform health informatics development and policies in the years to come. This systematic review aims to evaluate the impact of EHR interoperability on patient safety in health systems of high-income countries.

**Methods and analysis** A systematic literature review will be conducted via a computerised search through four databases: PubMed, Embase, Health Management Information Consortium and PsycInfo for relevant articles published between 2010 and 2020. Outcomes of interest will include impact on patient safety and the broader effects on health systems. Quality of the randomised quantitative studies will be assessed using Cochrane Risk of Bias Tool. Non-randomised papers will be evaluated with the Risk of Bias In Non-Randomised Studies—of Interventions tool. Drummond's Checklist will be used for publications pertaining to economic evaluation. The National Institute for Health and Care Excellence quality appraisal checklist will be used to assess qualitative studies. A narrative synthesis will be conducted for included studies, and the body of evidence will be summarised in a summary of findings table.

**Ethics and dissemination** This review will summarise published studies with non-identifiable data and, thus, does not require ethical approval. Findings will be disseminated through preprints, open access peer-reviewed publications, and conference presentations.

**PROSPERO registration number** CRD42020209285.

## Strengths and limitations of this study

► Inclusion of quantitative, qualitative and mixed-methods studies can provide a comprehensive overview of the multitude of ways in which interoperable electronic health records (EHRs) may affect patient safety and health systems.

► Using robust methodology to examine the wealth of existing literature, the proposed systematic review attempts to answer a pragmatic question that is integral to future health informatics development and policies.

► The heterogeneity of methods and outcomes assessed may potentially obscure the true effect interoperable EHRs may have had on patient safety.

► Potential small sample size in subgroup analyses may negatively impact the statistical power in quantitative data synthesis.

► Limiting the search strategy to English-only publications may not capture studies exploring EHR experiences in non-English-speaking countries.

## INTRODUCTION

Electronic health records (EHRs) have become an integral part of modern healthcare since their initial mainstream implementation in the mid-late 2000s through the passing of the Health Information Technology (HIT) for Economic and Clinical Health Act in the US and the National Health Service (NHS) National Programme for IT initiative (NPfIT) in England.[1–4] From the documentation and retrieval of patient records and the prescription of medications, to coordinating complex care plans between different healthcare providers and electronic billing, EHRs fulfil a multitude of roles for both clinicians and patients alike.[5–9]

In order to achieve EHR's full potential, it is critical to improve interoperability—that is, *'the ability of health information systems to work together within and across organisation boundaries in order to advance effective delivery of healthcare for individuals and communities'*.[10] The lack of universal interoperability is often cited as one of the many significant shortcomings of

**BMJ**

EHRs currently in use, resulting in duplication in healthcare costs, increased clinician workload fatigue and poses a potential risk to patient safety.[2] This is especially problematic for patient populations with chronic conditions, polypharmacy and multiple comorbidities who are reliant on effective patient information sharing via EHRs to facilitate their care.[11]

Poor EHR interoperability is detrimental to patient safety and costly for health systems. Its consequences range from increased risks of medication errors, fragmentation of patient data, to iatrogenic harm resulting from redundant testing, and additional healthcare expenditure.[12–17] In the fragmented EHR landscape of the UK, measuring the effect of poor interoperability remains challenging.[18] Although there is a growing body of literature investigating areas such as the facilitators and barriers to EHR greater adoption, technical capabilities, and usability,[19 20] no systematic review has been conducted exploring specifically the problem of interoperability among the assortment of EHRs in use, how it affects patient safety, and ultimately the financial cost savings lost to health systems.

In a recent systematic review by Dobrow *et al* assessing the effects of EHR and HIT interoperability on health systems, 130 publications were included, with the majority being studies conducted in the US, used quantitative methods and focused primarily on acute healthcare settings. The authors noted that the use of interoperable EHRs had a positive impact on outcome measures such as quality of care and productivity.[19] However, in domains such as stakeholder engagement, performance and reliability, security and privacy, information quality and ease of use, the benefits of interoperable EHRs were less clear.[19] Among the 130 publications, 17 were reviews with the majority directed at exploring facilitators and barriers to EHR implementation and the general benefits and impact of EHR use. While this review did focus on studies pertaining to the topic of interoperable EHRs, this was done from a broad perspective and included studies exploring a wide range of outcomes related to the effects of EHR on healthcare rather than specifically on their implications to patient safety.

In another review by Hersh *et al*, the authors explored how health information exchange (HIE) affected health systems on a variety of domains, including costs, healthcare utilisation, health outcomes, healthcare worker attitudes and sustainability. Despite the widespread routinely use of HIE, the authors described a general lack of robust evidence on the quality, costs, efficiency, usage and sustainability.[21] However, there was some evidence demonstrating HIEs being associated with reduced utilisation and costs in emergency care settings despite methodological issues being present in many of the included publications.[21] Although this review was ambitious in the wide scope of interest regarding the effects of HIE use, patient safety was not a primary topic of focus. Another limitation of this study was that it only contained US-based publications, and, thus, findings lack generalisability

internationally to other health systems in high-income countries (HIC) which are both organised and financed differently.

## RESEARCH AIM

The overall aim of this literature review is to explore how EHR interoperability impacts patient safety, in the context of health systems in HICs. The results generated will aim to inform healthcare policymakers and help shape more effective EHR system implementation and modernisation efforts in the coming years.

## METHODS AND ANALYSIS
### Search strategy

A computerised search of the literature published in the last 10 years (2010–2020) will be performed on PubMed/Medline, Embase, Cumulative Index to Nursing and Allied Health Literature, Health Management Information Consortium and PsycInfo. This publication timeframe was chosen as it coincides with the mainstream implementation of EHRs in several HIC healthcare systems such as Kaiser Permanente in the US, and, thus, would select for the most up to date, relevant evidence concerning EHR interoperability and patient safety challenges faced by healthcare systems today to be included.[22 23] The list of search strings used will include both free text and controlled terms, whenever supported (table 1) and will be iteratively refined in consultation with the Imperial College St. Mary's campus medical librarian. For a sample of the search strategy, please see online supplemental file 1.

Grey literature sources will also be searched, including registrations in the International Prospective Register of Systematic Reviews, reports of relevant stakeholder organisations (NHS England, American Medical Informatics Association (AMIA), eHealth at WHO and conference proceedings (last five years) of several related conferences (AMIA, MedInfo, Medicine V.2.0, Medicine X)), in order to identify possible additional studies that meet the inclusion criteria.

The search has also been restricted to HIC and articles published in English only.

### Study selection criteria

A summary of the population, intervention, comparison, outcomes and type of studies being considered is provided in table 2. This systematic review will focus on studies performed in HIC and published in English only. HIC will be defined in accordance with the World Bank's definition of *'countries where the gross national income (GNI) per capita is higher than $12 536 USD'*.[24] Studies assessing the impact of EHR interoperability will be included. Interventions will include EHR systems interoperable with other HIT systems both within and across healthcare facilities as well as those used in tertiary and community settings. The primary outcomes to be considered in this review will

**Table 1** Concepts and database search terms

| Electronic health records | | Interoperability | | Patient safety |
|---|---|---|---|---|
| ► Electronic health records<br>► Electronic medical records<br>► Computerised medical records systems<br>► Health information exchange<br>► Health information technology<br>► Hospital information systems<br>► Medical informatics<br>► Medical records linkage | AND | ► Interoperability<br>► Health information interoperability<br>► Systems integration | AND | ► Patient safety<br>► Patient adj1 incident*<br>► Adverse adj1 event*<br>► Patient adj1 outcome*<br>► Patient adj1 harm<br>► Risk management |

The asterisk is a truncation, a search method which will return all iterations/derivations of the term being queried (e.g., book* will return search results containing, book, books, booklet, booked etc).

be safety outcomes, including adverse events/incidents, safety-related patient experiences and health outcomes. In addition, secondary outcomes would include studies exploring the broader impact of interoperable EHRs on health systems such as cost-effectiveness and clinical culture among healthcare providers on the topics quantitative, qualitative, and mixed methods studies. Reference lists of the selected articles will also be screened for papers that may have been missed by the initial database search but still meet the eligibility criteria.

## Screening

Articles to be included will be screened by two independent reviewers, following the process described in the Preferred Reporting Items for Systematic Reviews and Meta-Analyses (PRISMA) flow diagram.[25] The initial screening will be done by the first reviewer based on the publication titles, followed by a second screening based on the abstracts. Included abstracts will then be fully reviewed by two independent researchers to produce a unified selection of articles to be included in this review. Cohen's kappa will be calculated to ensure inter-rater agreement and consistency in the selection of studies to be included.[26 27] Any disagreements will be resolved by consensus; if a Cohen's kappa value of less than 0.6 is reported, the discrepancies will be addressed through discussions with a more experienced third investigator.

## Data extraction

Data extraction will be performed using a standardised extraction table for each of the two investigators to summarise the characteristics and findings of each

**Table 2** PICO inclusion criteria

| | |
|---|---|
| **Population** | High-income countries using electronic health records |
| **Intervention** | EHRs with interoperability |
| **Comparison** | Usual care (i.e., existing baseline of interoperability) |
| **Outcome** | Impact on patient safety and quality of care |

EHRs, electronic health records.

included study, including name of the first author, year of publication, study design, number of participants, retention rates, setting characteristics, outcome measures and main results. The content of the two summary tables will then be aggregated and reviewed once more by both investigators, with any disagreements being solved by the third senior investigator.

## Quality assessment

The quality of randomised controlled trials and cluster randomised trials will be assessed using the Cochrane Risk of Bias Tool,[28] and the quality of nonrandomised intervention studies (i.e., case–control, cohort, quasi-experimental) will be appraised using the 'Risk of Bias In Non-Randomised Studies—of Interventions' tool.[29] For cost-effectiveness studies, the Drummond's checklist for assessing economic evaluations will be used.[30] The National Institute for Health and Care Excellence quality appraisal checklist will be used to assess the selected qualitative studies.[31] Two independent reviewers will score the selected studies and any disagreements will be resolved by a third person. A risk of bias table along with an overall, collective bias narrative will be produced to summarise the biases of outcomes observed among the evaluated studies.

## Narrative synthesis, subgroup analysis and meta-analysis

A narrative synthesis will be performed for all studies included in this systematic review to summarise any salient findings observed.[32]

In quantitative studies with homogenous or comparable outcome measures, whenever possible, continuous and dichotomous outcomes will be pooled together in a meta-analysis. If possible, effect sizes will be transformed in a common metric (Hedges' g—the bias-corrected standardised difference in means) and classified as positive when in favour of the intervention. Heterogeneity will be assessed using $I^2$ and the presence of publication bias will be evaluated using a funnel plot and the Duval and Tweedie's trim and fill method.[33]

For both qualitative and quantitative studies that report comparable outcomes, a subgroup analysis based on clinical settings (e.g., primary vs secondary healthcare settings) will

be conducted to explore any patterns or relationships ascertained from the data. Through a standardised spreadsheet shared among the reviewers, the body of evidence will be organised in two separate Summary of Findings tables (for both qualitative and quantitative studies) in accordance to the 'Grading of Recommendations Assessment, Development and Evaluation' criteria.[34]

### Patient and public involvement

This systematic literature review saw no direct participation by patients or the public during the design of this study. However, this study was designed following a series of structured interviews with patients regarding their experience of attending multiple institutions for hospital care.[35] As this literature review will be used to form the basis for subsequent studies exploring the topic including ones involving patients, findings from this review will be shared with patient research groups to gain feedback and encourage further discourse surrounding the topic of EHR interoperability and patient safety.

### Amendments

Any amendments to this protocol will be documented with reference to saved searches and analysis methods, which will be recorded in bibliographic databases, Mendeley, and Excel templates for data collection and synthesis.

### DISCUSSION

One of the primary strengths stemming from the almost exploratory nature of this systematic review is the ability to generate a succinct, comprehensive appraisal of the best evidence currently available regarding how EHR interoperability impacts patient care and safety. By publishing this review protocol beforehand, we demonstrate a clear, robust, and transparent approach to aggregating the anticipated assortment of literature on the subject in question.

There are also some limitations to be acknowledged. By restricting the inclusion criteria to publications made in English only, this could potentially exclude relevant papers pertaining to interoperable EHR systems in non-English healthcare settings. However, this is expected to be minimal as the majority of the papers concerning this topic published from the US and European countries and are primarily done so in English journals. It must also be noted that both the heterogeneity of measures and outcomes evaluated, as well as the potentially reduced number of studies in subgroup analyses, may negatively influence the statistical power in data synthesis and preclude the pooling of data to form a robust meta-analysis. With such diverse means of measuring and assessing the effects of EHR interoperability, this will likely make comparisons between studies difficult and may obscure the true measure of effect EHR interoperability has had in the clinical setting. To mitigate this risk, outcomes will be grouped whenever possible and summarised as a narrative synthesis. However, this can also

represent a strength, as it will provide a comprehensive overview on the subject, capitalising on various research methodologies and provide novel insights into the impact of interoperable EHR systems on patient safety.

### ETHICS AND DISSEMINATION

This review will summarise published studies with non-identifiable data and, therefore, does not require ethical approval. This protocol complies with the PRISMA Protocols guidelines. Findings will be disseminated through preprints, open access peer-reviewed publication, and conference presentations.

**Author affiliations**
[1]Patient Safety Translational Research Centre, Institute of Global Health Innovation, Department of Surgery & Cancer, Imperial College London, London, UK
[2]Centre for Mathematics of Precision Healthcare, Department of Mathematics, Imperial College London, London, UK
[3]Center for Health Technology and Services Research, Department of Community Medicine, Health Information and Decision, University of Porto, Porto, Portugal

**Acknowledgements** We would like to thank Michael Gainsford (Library Manager and Liaison Librarian at Imperial College London) for his support and guidance provided to improve the composition of the search terms and procedural aspects of the overall search strategy.

**Contributors** Conception and design of the work: EL, ALN and JC wrote the manuscript. HA and AD provided critical revision of drafts for important intellectual content. All authors provided input into drafts of the manuscript and agree on the contents of the final version.

**Funding** This research was supported through the Imperial College National Institute for Health Research (NIHR) Patient Safety Translational Research Centre (PSTRC) and the Imperial College Biomedical Research Centre (BRC). JC acknowledges support from the Wellcome Trust [215938/Z/19/Z]. However, the funder/sponsor has had no role in development and drafting of this protocol.

**Competing interests** None declared.

**Patient consent for publication** Not required.

**Provenance and peer review** Not commissioned; externally peer reviewed.

**ORCID iDs**
Edmond Li http://orcid.org/0000-0001-8209-4490
Jonathan Clarke http://orcid.org/0000-0003-1495-7746
Ana Luisa Neves http://orcid.org/0000-0002-7107-7211
Hutan Ashrafian http://orcid.org/0000-0003-1668-0672

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
