## [Reviewer comments · BMJ Open]

ARTICLE DETAILS

TITLE (PROVISIONAL)	Electronic Health Records, Interoperability, and Patient Safety in Health Systems of High-Income Countries: A Systematic Review Protocol
AUTHORS	Li, Edmond; Clarke, Jonathan; Neves, Ana Luisa; Ashrafian, Hutan; Darzi, Ara

VERSION 1 – REVIEW

REVIEWER	Meg E. Morris La Trobe University Australia
REVIEW RETURNED	20-Oct-2020

GENERAL COMMENTS	Clearly written and well thought through protocol on a highly clinically relevant topic. I only have 2 questions: (1) why was the database search restricted to such a small number of databases? Please consider expanding to include CINHAL and some of the other computer sciences and quality data bases. (2) Please provide further details on the exact strategies for statistical analysis.
--

REVIEWER	Marie-Pierre Gagnon Université Laval, Canada
REVIEW RETURNED	09-Dec-2020

GENERAL COMMENTS	This manuscript presents the protocol for a systematic review on electronic health record (EHR) interoperability on patient safety. It adheres to the Preferred Reporting Items for Systematic Review and Meta-Analyses Protocols (PRISMA-P) guidelines and describes the methods that are planned to conduct the systematic review. Overall, the protocol is clearly presented and the proposed methods are sound. There are only a couple of points that would need more explanation: 1. The decision to limit the outcomes of interest to safety outcomes could be better presented in the Introduction.2. The 10-year limit regarding date of publication (2010-2020) should be justified since interoperable EHRs were already present before 2010.3. Excluding studies published in other languages than English is a limitation that should be addressed.4. Both qualitative and quantitative studies will be included. What about mixed-methods studies?
---

	5. A reference for the thematic narrative that will be employed for synthesising qualitative studies should be provided. 6. Are there any sub-group analysis planned, for instance by comparing studies conducted in different regions or targeting different populations or settings of care? 7. In the Discussion, the limitation regarding the anticipated heterogeneity in methodology of the included articles which would preclude a quantitative synthesis is not based on empirical evidence... What if the authors find a cluster of studies that are more homogeneous and could be combined in a meta-analysis?
--	--

VERSION 1 – AUTHOR RESPONSE

Reviewer: 1

Comments to the Author

Clearly written and well thought through protocol on a highly clinically relevant topic. I only have 2 questions:

(1) Why was the database search restricted to such a small number of databases? Please consider expanding to include CINHAL and some of the other computer sciences and quality data bases.

Thank you for your comment. We have made changes to include the CINAHL and PsycInfo databases now in our search. The manuscript currently reads as follows (lines 160-165):

“A computerised search of the literature published in the last 10 years (March 2010 to March 2020) search will be performed on PubMed/Medline, Embase, Cumulative Index to Nursing and Allied Health Literature (CINAHL), Health Management Information Consortium (HMIC) and PsycInfo.”

(2) Please provide further details on the exact strategies for statistical analysis.

Thank you for your comment. We have substantially changed the manuscript to improve the section “Narrative synthesis, subgroups analysis, and meta-analysis” in the methods, in order to improve clarity on the data analysis. In brief, data analysis will include 1) a narrative synthesis of all studies included, 2) if possible and depending on data characteristics, comparable quantitative measures will be pooled together in a meta-analysis, and 3) a subgroup analysis based on clinical setting.

In the current version we also expanded the statistical details of the potential meta-analysis, explicitly mentioning the use of a common metric (Hedges’ g – the bias-corrected standardised difference in means), heterogeneity assessment using I^2 , and evaluation of the presence of publication bias using the funnel plot and the Duval and Tweedie trim and fill method.

As a result, this amended section currently reads as follows (lines 225-241):

“A narrative synthesis will be performed for all studies included in this systematic review to summarise any salient findings observed (29).

In quantitative studies with homogenous or comparable outcome measures, whenever possible, continuous and dichotomous outcomes will be pooled together in a meta-analysis. If possible, effect sizes will be transformed in a common metric (Hedges’ g – the bias-corrected standardised difference in means) and classified as positive when in favour of the intervention. Heterogeneity will be assessed using I^2 and the presence of publication bias will be evaluated using a funnel plot and the Duval and Tweedie’s trim and fill method (30).

For both qualitative and quantitative studies that reporting comparable outcomes, a subgroup analysis based on clinical settings (e.g., primary vs. secondary healthcare settings) will be conducted to explore any patterns or relationships ascertained from the data. Through a standardised spreadsheet shared amongst the reviewers, the body of evidence will be organised in two separate Summary of Findings tables (for both qualitative and quantitative studies) in accordance to the 'Grading of Recommendations Assessment, Development and Evaluation' (GRADE) criteria (31)."

However, we are aware that the characteristics and quality of the data might be a limiting factor on the type of analysis performed. Therefore, we also added a paragraph about expected limitations of this study, including the fact that the heterogeneity of measures and outcomes evaluated, as well as potentially reduced number of studies in subgroup analyses, may negatively influence the statistical power in data synthesis, and not allow data pooling in a meta-analysis. The corresponding section of the manuscript currently reads as follows (lines 267-270):

"It must also be noted that both heterogeneity of measures and outcomes evaluated, as well as potentially reduced number of studies in subgroup analyses, may negatively influence the statistical power in data synthesis, and may preclude pooling of data as a meta-analysis."

Reviewer: 2

Comments to the Author

This manuscript presents the protocol for a systematic review on electronic health record (EHR) interoperability on patient safety. It adheres to the Preferred Reporting Items for Systematic Review and Meta-Analyses Protocols (PRISMA-P) guidelines and describes the methods that are planned to conduct the systematic review. Overall, the protocol is clearly presented and the proposed methods are sound.

There are only a couple of points that would need more explanation:

1. The decision to limit the outcomes of interest to safety outcomes could be better presented in the Introduction.

Thank you for this comment. We agree this could have been better presented. We have made changes to the manuscript to clarify the importance of the relationship between EHR interoperability, patient safety and health systems. The edited section currently reads as follows (lines 118-122):

"Poor EHR interoperability is detrimental to patient safety and costly for health systems. Its consequences range from increased risks of medication errors, fragmentation of patient data, to iatrogenic harm resulting from redundant testing, and additional healthcare expenditure (12–17). In the fragmented EHR landscape of the United Kingdom, measuring the effect poor EHR interoperability has in the National Health Service (NHS), remain challenging (18)."

2. The 10-year limit regarding date of publication (2010-2020) should be justified since interoperable EHRs were already present before 2010.

Thank you for this comment. While we acknowledge that EHR interoperability has been a longstanding issue since the introduction of this technology, we believe publications and their findings dating back longer than 10 years ago may obscure more recent publications which are more relevant to the current health informatics landscape. The 10-year date of publication limit was chosen to select more recent papers which explore the topic from a more up to date perspective. This 10 year limit was also present in similar systematic reviews exploring other EHR topics as well [2].

This has been added in the manuscript, that currently reads as follows (lines 163-167):

“This publication timeframe was chosen as it coincides with the mainstream implementation of EHRs in several HIC healthcare systems such as Kaiser Permanente in the US, and thus would select for the most up to date, relevant evidence concerning EHR interoperability and patient safety challenges faced by healthcare systems today to be included (22,23).”

3. Excluding studies published in other languages than English is a limitation that should be addressed.

Thank you for your comments on this. Limiting studies to those only published in English would indeed exclude some papers and findings derived from non-English settings. However, this is expected to be minimal as the majority of the papers concerning this topic published from the United States and European countries and are primarily done so in English journals. It should be noted that some relevant EHR publications from non-English speaking areas of the world have also been captured despite this criteria [3]. This limitation to English-only publications is also common to systematic reviews exploring other EHR topics as well [2,4]. Hence, this is a recognised and accepted limitation of this review nonetheless.

We have amended the manuscript text to further clarify this limitation, currently reading as follows (lines 263-267):

“By restricting the inclusion criteria to publications made English only, this could potentially exclude relevant papers pertaining to interoperable EHR systems in non-English healthcare settings. However, this is expected to be minimal as the majority of the papers concerning this topic published from the United States and European countries and are primarily done so in English journals.”

4. Both qualitative and quantitative studies will be included. What about mixed-methods studies?

Thank you for highlighting this. We agree with it and have incorporated mixed-methods papers into our review now. Changes have been made in the manuscript to reflect this (lines 79, 188-189):

“Quantitative, qualitative, and mixed methods studies will be included.”

5. A reference for the thematic narrative that will be employed for synthesising qualitative studies should be provided.

Thank you for your comment on this. We agree that the way it was presented in the initial draft of this protocol manuscript was confusing. We have now rectified this to clarify that we are doing a narrative synthesis instead for all papers included in this review. Amendments have been made to the text and an appropriate reference has been included. (lines 225-226) [5]

6. Are there any sub-group analysis planned, for instance by comparing studies conducted in different regions or targeting different populations or settings of care?

Thank you for your comment on this. Our ability to have a subgroup analysis done would ultimately depend on the findings from the selected literature to be included in this review. Should it be possible, subgroup analyses based on care settings will be conducted. Changes have been made in the manuscript to clarify this now (lines 235-237):

“For both qualitative and quantitative studies that reporting comparable outcomes, a subgroup analysis based on clinical settings (e.g., primary vs. secondary healthcare settings) will be conducted to explore any patterns or relationships ascertained from the data.”

7. In the Discussion, the limitation regarding the anticipated heterogeneity in methodology of the included articles which would preclude a quantitative synthesis is not based on empirical evidence... What if the authors find a cluster of studies that are more homogeneous and could be combined in a meta-analysis?

Thank you for pointing this out. We have edited the manuscript in order to clearly describe the strategy planned in case a cluster of homogeneous studies are found, and therefore we have an opportunity to pool comparable data in a meta-analysis. The edited manuscript currently reads as follows (lines 228-233):

“In quantitative studies with homogenous or comparable outcome measures, whenever possible, continuous and dichotomous outcomes will be pooled together in a meta-analysis. If possible, effect sizes will be transformed in a common metric (Hedges’ g – the bias-corrected standardised difference in means) and classified as positive when in favour of the intervention. Heterogeneity will be assessed using I^2 and the presence of publication bias will be evaluated using a funnel plot and the Duval and Tweedie’s trim and fill method. [1]”

REFERENCES

- 1 Duval S, Tweedie R. Trim and fill: A simple funnel-plot-based method of testing and adjusting for publication bias in meta-analysis. *Biometrics* 2000;**56**:455–63. doi:10.1111/j.0006-341X.2000.00455.x
- 2 Dobrow MJ, Bytautas JP, Tharmalingam S, *et al.* Interoperable Electronic Health Records and Health Information Exchanges: Systematic Review. *JMIR Med Informatics* 2019;**7**:e12607. doi:10.2196/12607
- 3 Alotaibi YK, Federico F. The impact of health information technology on patient safety. *Saudi Med J* 2017;**38**:1173–80. doi:10.15537/smj.2017.12.20631
- 4 Kruse CS, Kristof C, Jones B, *et al.* Barriers to Electronic Health Record Adoption: a Systematic Literature Review. *J Med Syst* 2016;**40**. doi:10.1007/s10916-016-0628-9
- 5 Popay, J., Roberts, H., Sowden, A., Petticrew, M., Arai, L., Rodgers, M., ... & Duffy S. Guidance on the conduct of narrative synthesis in systematic reviews. A product from the ESRC methods programme. 2006;:1–92.

VERSION 2 – REVIEW

REVIEWER	Morris, Meg La Trobe Univ, Room 423 HS3
REVIEW RETURNED	18-Feb-2021

GENERAL COMMENTS	Thank you for the very good revisions. My only feedback now is for the term "high income counties" to be changed to other wording as it is not clear exactly what this means: eg "This systematic review aims to evaluate the impact of EHR interoperability on patient safety in health systems of high-income countries."
---

REVIEWER	Gagnon, Marie-Pierre University of Laval
REVIEW RETURNED	25-Feb-2021

VERSION 2 – AUTHOR RESPONSE**Reviewer: 1****Prof. Meg Morris, La Trobe University****Comments to the Author:**

Thank you for the very good revisions. My only feedback now is for the term "high income countries" to be changed to other wording as it is not clear exactly what this means: e.g. "This systematic review aims to evaluate the impact of EHR interoperability on patient safety in health systems of high-income countries."

Thank you for your comment. We used this terminology in the manuscript as this is the current terminology used by the World Bank [1]. We acknowledge that while there were other terms previously used that may be more explicit (e.g., developed countries), these are not used anymore as they are no longer deemed appropriate by the World Bank [2]. To address the reviewer's concerns, we have added a definition of high-income country in the manuscript for greater clarity (lines 177-179):

High-income countries will be defined in accordance with the World Bank's definition of "countries where the gross national income (GNI) per capita is higher than \$12,536 USD"

Reviewer: 2**Dr. Marie-Pierre Gagnon, University of Laval****Comments to the Author:**

Thank you for addressing all my comments!

Thank you for your feedback and taking the time to review our work.

References

- 1 World Bank. World Bank Country and Lending Groups – World Bank Data Help Desk. World Bank. 2019;:1–8.<https://datahelpdesk.worldbank.org/knowledgebase/articles/906519> (accessed 26 May 2021).
- 2 World Bank. Should we continue to use the term “developing world”? 2015;:1–8.<https://blogs.worldbank.org/opendata/should-we-continue-use-term-developing-world> (accessed 26 May 2021).